# MLP-Mixer: An all-MLP Architecture for Vision

**Ilya Tolstikhin**[*], **Neil Houlsby**[*], **Alexander Kolesnikov**[*], **Lucas Beyer**[*],

**Xiaohua Zhai, Thomas Unterthiner, Jessica Yung, Andreas Steiner,**

**Daniel Keysers, Jakob Uszkoreit, Mario Lucic, Alexey Dosovitskiy**

[*]equal contribution

Google Research, Brain Team

```
{tolstikhin, neilhoulsby, akolesnikov, lbeyer,
 xzhai, unterthiner, jessicayung†, andstein,
keysers, usz, lucic, adosovitskiy}@google.com
```

[†]work done during Google AI Residency

## Abstract

Convolutional Neural Networks (CNNs) are the go-to model for computer vision. Recently, attention-based networks, such as the Vision Transformer, have also become popular. In this paper we show that while convolutions and attention are both sufficient for good performance, neither of them are necessary. We present *MLP-Mixer*, an architecture based exclusively on multi-layer perceptrons (MLPs). MLP-Mixer contains two types of layers: one with MLPs applied independently to image patches (i.e. "mixing" the per-location features), and one with MLPs applied across patches (i.e. "mixing" spatial information). When trained on large datasets, or with modern regularization schemes, MLP-Mixer attains competitive scores on image classification benchmarks, with pre-training and inference cost comparable to state-of-the-art models. We hope that these results spark further research beyond the realms of well established CNNs and Transformers.[1]

## 1 Introduction

As the history of computer vision demonstrates, the availability of larger datasets coupled with increased computational capacity often leads to a paradigm shift. While Convolutional Neural Networks (CNNs) have been the de-facto standard for computer vision, recently Vision Transformers [14] (ViT), an alternative based on self-attention layers, attained state-of-the-art performance. ViT continues the long-lasting trend of removing hand-crafted visual features and inductive biases from models and relies further on learning from raw data.

We propose the *MLP-Mixer* architecture (or "Mixer" for short), a competitive but conceptually and technically simple alternative, that does not use convolutions or self-attention. Instead, Mixer's architecture is based entirely on multi-layer perceptrons (MLPs) that are repeatedly applied across either spatial locations or feature channels. Mixer relies only on basic matrix multiplication routines, changes to data layout (reshapes and transpositions), and scalar nonlinearities.

Figure 1 depicts the macro-structure of Mixer. It accepts a sequence of linearly projected image patches (also referred to as *tokens*) shaped as a "patches × channels" table as an input, and maintains this dimensionality. Mixer makes use of two types of MLP layers: *channel-mixing MLPs* and *token-mixing MLPs*. The channel-mixing MLPs allow communication between different channels;

---

[1]MLP-Mixer code is available at https://github.com/google-research/vision_transformer

35th Conference on Neural Information Processing Systems (NeurIPS 2021).

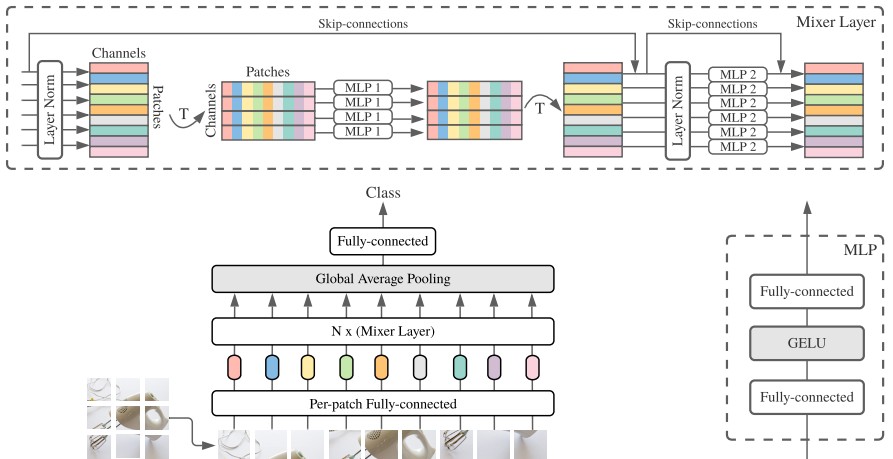

Figure 1: MLP-Mixer consists of per-patch linear embeddings, Mixer layers, and a classifier head. Mixer layers contain one token-mixing MLP and one channel-mixing MLP, each consisting of two fully-connected layers and a GELU nonlinearity. Other components include: skip-connections, dropout, and layer norm on the channels.

they operate on each token independently and take individual rows of the table as inputs. The token-mixing MLPs allow communication between different spatial locations (tokens); they operate on each channel independently and take individual columns of the table as inputs. These two types of layers are interleaved to enable interaction of both input dimensions.

In the extreme case, our architecture can be seen as a very special CNN, which uses $1{\times}1$ convolutions for *channel mixing*, and single-channel depth-wise convolutions of a full receptive field and parameter sharing for *token mixing*. However, the converse is not true as typical CNNs are not special cases of Mixer. Furthermore, a convolution is more complex than the plain matrix multiplication in MLPs as it requires an additional costly reduction to matrix multiplication and/or specialized implementation.

Despite its simplicity, Mixer attains competitive results. When pre-trained on large datasets (i.e., ~100M images), it reaches near state-of-the-art performance, previously claimed by CNNs and Transformers, in terms of the accuracy/cost trade-off. This includes 87.94% top-1 validation accuracy on ILSVRC2012 "ImageNet" [13]. When pre-trained on data of more modest scale (i.e., ~1–10M images), coupled with modern regularization techniques [49, 54], Mixer also achieves strong performance. However, similar to ViT, it falls slightly short of specialized CNN architectures.

## 2 Mixer Architecture

Modern deep vision architectures consist of layers that mix features (i) at a given spatial location, (ii) between different spatial locations, or both at once. In CNNs, (ii) is implemented with $N \times N$ convolutions (for $N > 1$) and pooling. Neurons in deeper layers have a larger receptive field [1, 29]. At the same time, $1{\times}1$ convolutions also perform (i), and larger kernels perform both (i) and (ii). In Vision Transformers and other attention-based architectures, self-attention layers allow both (i) and (ii) and the MLP-blocks perform (i). The idea behind the Mixer architecture is to clearly separate the per-location (*channel-mixing*) operations (i) and cross-location (*token-mixing*) operations (ii). Both operations are implemented with MLPs. Figure 1 summarizes the architecture.

Mixer takes as input a sequence of $S$ non-overlapping image patches, each one projected to a desired *hidden dimension* $C$. This results in a two-dimensional real-valued input table, $\mathbf{X} \in \mathbb{R}^{S \times C}$. If the original input image has resolution $(H, W)$, and each patch has resolution $(P, P)$, then the number of patches is $S = HW/P^2$. All patches are linearly projected with the *same* projection matrix. Mixer consists of multiple layers of identical size, and each layer consists of two MLP blocks. The first one is the *token-mixing* MLP: it acts on columns of $\mathbf{X}$ (i.e. it is applied to a transposed input table $\mathbf{X}^\top$), maps $\mathbb{R}^S \mapsto \mathbb{R}^S$, and is shared across all columns. The second one is the *channel-mixing* MLP: it acts on rows of $\mathbf{X}$, maps $\mathbb{R}^C \mapsto \mathbb{R}^C$, and is shared across all rows. Each MLP block contains two

fully-connected layers and a nonlinearity applied independently to each row of its input data tensor. Mixer layers can be written as follows (omitting layer indices):

$$\mathbf{U}_{*,i} = \mathbf{X}_{*,i} + \mathbf{W}_2\,\sigma\big(\mathbf{W}_1\,\mathrm{LayerNorm}(\mathbf{X})_{*,i}\big), \quad \text{for } i = 1 \ldots C, \tag{1}$$

$$\mathbf{Y}_{j,*} = \mathbf{U}_{j,*} + \mathbf{W}_4\,\sigma\big(\mathbf{W}_3\,\mathrm{LayerNorm}(\mathbf{U})_{j,*}\big), \quad \text{for } j = 1 \ldots S.$$

Here $\sigma$ is an element-wise nonlinearity (GELU [16]). $D_S$ and $D_C$ are tunable hidden widths in the token-mixing and channel-mixing MLPs, respectively. Note that $D_S$ is selected independently of the number of input patches. Therefore, the computational complexity of the network is linear in the number of input patches, unlike ViT whose complexity is quadratic. Since $D_C$ is independent of the patch size, the overall complexity is linear in the number of pixels in the image, as for a typical CNN.

As mentioned above, the *same* channel-mixing MLP (token-mixing MLP) is applied to every row (column) of $\mathbf{X}$. Tying the parameters of the channel-mixing MLPs (within each layer) is a natural choice—it provides positional invariance, a prominent feature of convolutions. However, tying parameters across channels is much less common. For example, separable convolutions [9, 40], used in some CNNs, apply convolutions to each channel independently of the other channels. However, in separable convolutions, a different convolutional kernel is applied to each channel unlike the token-mixing MLPs in Mixer that share the same kernel (of full receptive field) for all of the channels. The parameter tying prevents the architecture from growing too fast when increasing the hidden dimension $C$ or the sequence length $S$ and leads to significant memory savings. Surprisingly, this choice does not affect the empirical performance, see Supplementary A.1.

Each layer in Mixer (except for the initial patch projection layer) takes an input of the same size. This "isotropic" design is most similar to Transformers, or deep RNNs in other domains, that also use a fixed width. This is unlike most CNNs, which have a *pyramidal* structure: deeper layers have a lower resolution input, but more channels. Note that while these are the typical designs, other combinations exist, such as isotropic ResNets [38] and pyramidal ViTs [52].

Aside from the MLP layers, Mixer uses other standard architectural components: skip-connections [15] and layer normalization [2]. Unlike ViTs, Mixer does not use position embeddings because the token-mixing MLPs are sensitive to the order of the input tokens. Finally, Mixer uses a standard classification head with the global average pooling layer followed by a linear classifier. Overall, the architecture can be written compactly in JAX/Flax, the code is given in Supplementary F.

## 3 Experiments

We evaluate the performance of MLP-Mixer models, pre-trained with medium- to large-scale datasets, on a range of small and mid-sized downstream classification tasks. We are interested in three primary quantities: (1) Accuracy on the downstream task; (2) *Total* computational cost of pre-training, which is important when training the model from scratch on the upstream dataset; (3) Test-time throughput, which is important to the practitioner. Our goal is not to demonstrate state-of-the-art results, but to show that, remarkably, a simple MLP-based model is competitive with today's best convolutional and attention-based models.

**Downstream tasks** We use popular downstream tasks such as ILSVRC2012 "ImageNet" (1.3M training examples, 1k classes) with the original validation labels [13] and cleaned-up ReaL labels [5], CIFAR-10/100 (50k examples, 10/100 classes) [23], Oxford-IIIT Pets (3.7k examples, 36 classes) [33], and Oxford Flowers-102 (2k examples, 102 classes) [32]. We also use the Visual Task Adaptation Benchmark (VTAB-1k), which consists of 19 diverse datasets, each with 1k training examples [58].

**Pre-training** We follow the standard transfer learning setup: pre-training followed by fine-tuning on the downstream tasks. We pre-train our models on two public datasets: ILSVRC2021 ImageNet, and ImageNet-21k, a superset of ILSVRC2012 that contains 21k classes and 14M images [13]. To assess performance at larger scale, we also train on JFT-300M, a proprietary dataset with 300M examples and 18k classes [44]. We de-duplicate all pre-training datasets with respect to the test sets of the downstream tasks as done in Dosovitskiy et al. [14], Kolesnikov et al. [22]. We pre-train all models at resolution 224 using Adam with $\beta_1 = 0.9$, $\beta_2 = 0.999$, linear learning rate warmup of 10k steps and linear decay, batch size 4096, weight decay, and gradient clipping at global norm 1. For JFT-300M, we pre-process images by applying the cropping technique from Szegedy et al. [45] in addition to random horizontal flipping. For ImageNet and ImageNet-21k, we employ additional data augmentation and regularization techniques. In particular, we use RandAugment [12], mixup [60],

Table 1: Specifications of the Mixer architectures. The "B", "L", and "H" (base, large, and huge) model scales follow Dosovitskiy et al. [14]. A brief notation "B/16" means the model of base scale with patches of resolution 16×16. The number of parameters is reported for an input resolution of 224 and does not include the weights of the classifier head.

| Specification | S/32 | S/16 | B/32 | B/16 | L/32 | L/16 | H/14 |
|---|---|---|---|---|---|---|---|
| Number of layers | 8 | 8 | 12 | 12 | 24 | 24 | 32 |
| Patch resolution $P \times P$ | 32×32 | 16×16 | 32×32 | 16×16 | 32×32 | 16×16 | 14×14 |
| Hidden size $C$ | 512 | 512 | 768 | 768 | 1024 | 1024 | 1280 |
| Sequence length $S$ | 49 | 196 | 49 | 196 | 49 | 196 | 256 |
| MLP dimension $D_C$ | 2048 | 2048 | 3072 | 3072 | 4096 | 4096 | 5120 |
| MLP dimension $D_S$ | 256 | 256 | 384 | 384 | 512 | 512 | 640 |
| Parameters (M) | 19 | 18 | 60 | 59 | 206 | 207 | 431 |

dropout [43], and stochastic depth [19]. This set of techniques was inspired by the *timm library* [54] and Touvron et al. [48]. More details on these hyperparameters are provided in Supplementary B.

**Fine-tuning** We fine-tune using momentum SGD, batch size 512, gradient clipping at global norm 1, and a cosine learning rate schedule with a linear warmup. We do not use weight decay when fine-tuning. Following common practice [22, 48], we also fine-tune at higher resolutions with respect to those used during pre-training. Since we keep the patch resolution fixed, this increases the number of input patches (say from $S$ to $S'$) and thus requires modifying the shape of Mixer's token-mixing MLP blocks. Formally, the input in Eq. (1) is left-multiplied by a weight matrix $\mathbf{W}_1 \in \mathbb{R}^{D_S \times S}$ and this operation has to be adjusted when changing the input dimension $S$. For this, we increase the hidden layer width from $D_S$ to $D_{S'}$ in proportion to the number of patches and initialize the (now larger) weight matrix $\mathbf{W}_2' \in \mathbb{R}^{D_{S'} \times S'}$ with a block-diagonal matrix containing copies of $\mathbf{W}_2$ on its diagonal. This particular scheme only allows for $S' = K^2 S$ with $K \in \mathbb{N}$. See Supplementary C for further details. On the VTAB-1k benchmark we follow the BiT-HyperRule [22] and fine-tune Mixer models at resolution 224 and 448 on the datasets with small and large input images respectively.

**Metrics** We evaluate the trade-off between the model's computational cost and quality. For the former we compute two metrics: (1) Total pre-training time on TPU-v3 accelerators, which combines three relevant factors: the theoretical FLOPs for each training setup, the computational efficiency on the relevant training hardware, and the data efficiency. (2) Throughput in images/sec/core on TPU-v3. Since models of different sizes may benefit from different batch sizes, we sweep the batch sizes and report the highest throughput for each model. For model quality, we focus on top-1 downstream accuracy after fine-tuning. On two occasions (Figure 3, right and Figure 4), where fine-tuning all of the models is too costly, we report the few-shot accuracies obtained by solving the $\ell_2$-regularized linear regression problem between the frozen learned representations of images and the labels.

**Models** We compare various configurations of Mixer, summarized in Table 1, to the most recent, state-of-the-art, CNNs and attention-based models. In all the figures and tables, the MLP-based Mixer models are marked with pink (●), convolution-based models with yellow (●), and attention-based models with blue (●). The Vision Transformers (ViTs) have model scales and patch resolutions similar to Mixer. HaloNets are attention-based models that use a ResNet-like structure with local self-attention layers instead of 3×3 convolutions [51]. We focus on the particularly efficient "HaloNet-H4 (base 128, Conv-12)" model, which is a hybrid variant of the wider HaloNet-H4 architecture with some of the self-attention layers replaced by convolutions. Note, we mark HaloNets with both attention and convolutions with blue (●). Big Transfer (BiT) [22] models are ResNets optimized for transfer learning. NFNets [7] are normalizer-free ResNets with several optimizations for ImageNet classification. We consider the NFNet-F4+ model variant. We consider MPL [35] and ALIGN [21] for EfficientNet architectures. MPL is pre-trained at very large-scale on JFT-300M images, using meta-pseudo labelling from ImageNet instead of the original labels. We compare to the EfficientNet-B6-Wide model variant. ALIGN pre-train image encoder and language encoder on noisy web image text pairs in a contrastive way. We compare to their best EfficientNet-L2 image encoder.

### 3.1 Main results

Table 2 presents comparison of the largest Mixer models to state-of-the-art models from the literature. "ImNet" and "ReaL" columns refer to the original ImageNet validation [13] and cleaned-up ReaL [5]

Table 2: Transfer performance, inference throughput, and training cost. The rows are sorted by inference throughput (fifth column). Mixer has comparable transfer accuracy to state-of-the-art models with similar cost. The Mixer models are fine-tuned at resolution 448. Mixer performance numbers are averaged over three fine-tuning runs and standard deviations are smaller than $0.1$.

| | ImNet top-1 | ReaL top-1 | Avg 5 top-1 | VTAB-1k 19 tasks | Throughput img/sec/core | TPUv3 core-days |
|---|---|---|---|---|---|---|
| Pre-trained on ImageNet-21k (public) | | | | | | |
| ● HaloNet [51] | 85.8 | — | — | — | 120 | 0.10k |
| ● Mixer-L/16 | 84.15 | 87.86 | 93.91 | 74.95 | 105 | 0.41k |
| ● ViT-L/16 [14] | 85.30 | 88.62 | 94.39 | 72.72 | 32 | 0.18k |
| ● BiT-R152x4 [22] | 85.39 | — | 94.04 | 70.64 | 26 | 0.94k |
| Pre-trained on JFT-300M (proprietary) | | | | | | |
| ● NFNet-F4+ [7] | 89.2 | — | — | — | 46 | 1.86k |
| ● Mixer-H/14 | 87.94 | 90.18 | 95.71 | 75.33 | 40 | 1.01k |
| ● BiT-R152x4 [22] | 87.54 | 90.54 | 95.33 | 76.29 | 26 | 9.90k |
| ● ViT-H/14 [14] | 88.55 | 90.72 | 95.97 | 77.63 | 15 | 2.30k |
| Pre-trained on unlabelled or weakly labelled data (proprietary) | | | | | | |
| ● MPL [35] | 90.0 | 91.12 | — | — | — | 20.48k |
| ● ALIGN [21] | 88.64 | — | — | 79.99 | 15 | 14.82k |

labels. "Avg. 5" stands for the average performance across all five downstream tasks (ImageNet, CIFAR-10, CIFAR-100, Pets, Flowers). Figure 2 (left) visualizes the accuracy-compute frontier. When pre-trained on ImageNet-21k with additional regularization, Mixer achieves an overall strong performance (84.15% top-1 on ImageNet), although slightly inferior to other models[2]. Regularization in this scenario is necessary and Mixer overfits without it, which is consistent with similar observations for ViT [14]. The same conclusion holds when training Mixer from random initialization on ImageNet (see Section 3.2): Mixer-B/16 attains a reasonable score of 76.4% at resolution 224, but tends to overfit. This score is similar to a vanilla ResNet50, but behind state-of-the-art CNNs/hybrids for the ImageNet "from scratch" setting, e.g. 84.7% BotNet [42] and 86.5% NFNet [7].

When the size of the upstream dataset increases, Mixer's performance improves significantly. In particular, Mixer-H/14 achieves 87.94% top-1 accuracy on ImageNet, which is 0.5% better than BiT-ResNet152x4 and only 0.5% lower than ViT-H/14. Remarkably, Mixer-H/14 runs 2.5 times faster than ViT-H/14 and almost twice as fast as BiT. Overall, Figure 2 (left) supports our main claim that in terms of the accuracy-compute trade-off Mixer is competitive with more conventional neural network architectures. The figure also demonstrates a clear correlation between the total pre-training cost and the downstream accuracy, even across architecture classes.

BiT-ResNet152x4 in the table are pre-trained using SGD with momentum and a long schedule. Since Adam tends to converge faster, we complete the picture in Figure 2 (left) with the BiT-R200x3 model from Dosovitskiy et al. [14] pre-trained on JFT-300M using Adam. This ResNet has a slightly lower accuracy, but considerably lower pre-training compute. Finally, the results of smaller ViT-L/16 and Mixer-L/16 models are also reported in this figure.

## 3.2 The role of the model scale

The results outlined in the previous section focus on (large) models at the upper end of the compute spectrum. We now turn our attention to smaller Mixer models.

We may scale the model in two independent ways: (1) Increasing the model size (number of layers, hidden dimension, MLP widths) when pre-training; (2) Increasing the input image resolution when

---

[2]In Table 2 we consider the highest accuracy models in each class for each pre-training dataset. These all use the large resolutions (448 and above). However, fine-tuning at smaller resolution can lead to substantial improvements in the test-time throughput, with often only a small accuracy penalty. For instance, when pre-training on ImageNet-21k, the Mixer-L/16 model fine-tuned at 224 resolution achieves 82.84% ImageNet top-1 accuracy at throughput 420 img/sec/core; the ViT-L/16 model fine-tuned at 384 resolution achieves 85.15% at 80 img/sec/core [14]; and HaloNet fine-tuned at 384 resolution achieves 85.5% at 258 img/sec/core [51].

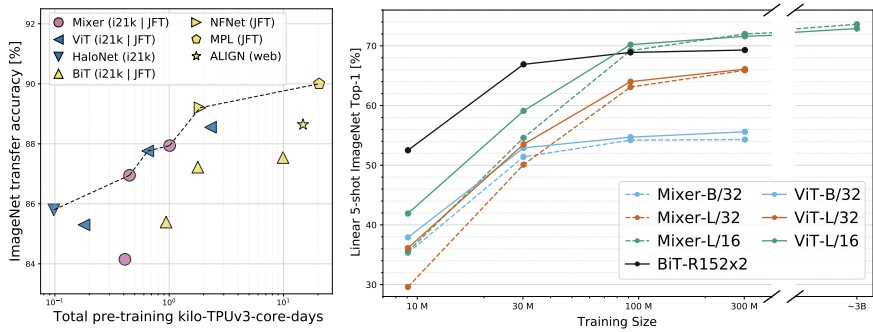

Figure 2: **Left:** ImageNet accuracy/training cost Pareto frontier (dashed line) for the SOTA models in Table 2. Models are pre-trained on ImageNet-21k, or JFT (labelled, or pseudo-labelled for MPL), or web image text pairs. Mixer is as good as these extremely performant ResNets, ViTs, and hybrid models, and sits on frontier with HaloNet, ViT, NFNet, and MPL. **Right:** Mixer (solid) catches or exceeds BiT (dotted) and ViT (dashed) as the data size grows. Every point on a curve uses the same pre-training compute; they correspond to pre-training on 3%, 10%, 30%, and 100% of JFT-300M for 233, 70, 23, and 7 epochs, respectively. Additional points at ∼3B correspond to pre-training on an even larger JFT-3B dataset for the same number of total steps. Mixer improves more rapidly with data than ResNets, or even ViT. The gap between large Mixer and ViT models shrinks.

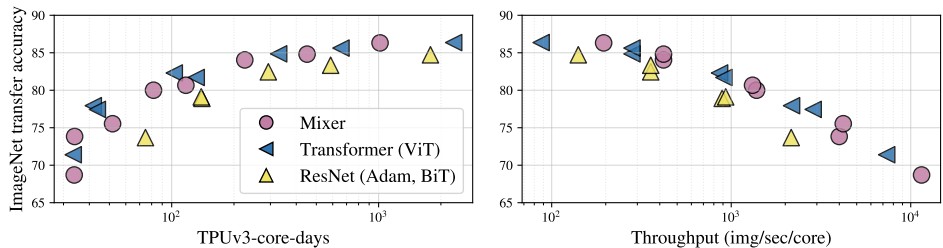

Figure 3: The role of the model scale. ImageNet validation top-1 accuracy vs. total pre-training compute (**left**) and throughput (**right**) of ViT, BiT, and Mixer models at various scales. All models are pre-trained on JFT-300M and fine-tuned at resolution 224, which is lower than in Figure 2 (left).

fine-tuning. While the former affects both pre-training compute and test-time throughput, the latter only affects the throughput. Unless stated otherwise, we fine-tune at resolution 224.

We compare various configurations of Mixer (see Table 1) to ViT models of similar scales and BiT models pre-trained with Adam. The results are summarized in Table 3 and Figure 3. When trained from scratch on ImageNet, Mixer-B/16 achieves a reasonable top-1 accuracy of 76.44%. This is 3% behind the ViT-B/16 model. The training curves (not reported) reveal that both models achieve very similar values of the training loss. In other words, Mixer-B/16 overfits more than ViT-B/16. For the Mixer-L/16 and ViT-L/16 models this difference is even more pronounced.

As the pre-training dataset grows, Mixer's performance steadily improves. Remarkably, Mixer-H/14 pre-trained on JFT-300M and fine-tuned at 224 resolution is only 0.3% behind ViT-H/14 on ImageNet whilst running 2.2 times faster. Figure 3 clearly demonstrates that although Mixer is slightly below the frontier on the lower end of model scales, it sits confidently on the frontier at the high end.

## 3.3 The role of the pre-training dataset size

The results presented thus far demonstrate that pre-training on larger datasets significantly improves Mixer's performance. Here, we study this effect in more detail.

To study Mixer's ability to make use of the growing number of training examples we pre-train Mixer-B/32, Mixer-L/32, and Mixer-L/16 models on random subsets of JFT-300M containing 3%, 10%, 30% and 100% of all the training examples for 233, 70, 23, and 7 epochs. Thus, every model is pre-trained for the same number of total steps. We also pre-train Mixer-L/16 model on an even larger JFT-3B dataset [59] containing roughly 3B images with 30k classes for the same number of total steps.

Table 3: Performance of Mixer and other models from the literature across various model and pre-training dataset scales. "Avg. 5" denotes the average performance across five downstream tasks. Mixer and ViT models are averaged over three fine-tuning runs, standard deviations are smaller than 0.15. (‡) Extrapolated from the numbers reported for the same models pre-trained on JFT-300M without extra regularization. (☎) Numbers provided by authors of Dosovitskiy et al. [14] through personal communication. Rows are sorted by throughput.

| | Image size | Pre-Train Epochs | ImNet top-1 | ReaL top-1 | Avg. 5 top-1 | Throughput (img/sec/core) | TPUv3 core-days |
|---|---|---|---|---|---|---|---|
| Pre-trained on ImageNet (with extra regularization) | | | | | | | |
| • Mixer-B/16 | 224 | 300 | 76.44 | 82.36 | 88.33 | 1384 | 0.01k[(‡)] |
| • ViT-B/16 (☎) | 224 | 300 | 79.67 | 84.97 | 90.79 | 861 | 0.02k[(‡)] |
| • Mixer-L/16 | 224 | 300 | 71.76 | 77.08 | 87.25 | 419 | 0.04k[(‡)] |
| • ViT-L/16 (☎) | 224 | 300 | 76.11 | 80.93 | 89.66 | 280 | 0.05k[(‡)] |
| Pre-trained on ImageNet-21k (with extra regularization) | | | | | | | |
| • Mixer-B/16 | 224 | 300 | 80.64 | 85.80 | 92.50 | 1384 | 0.15k[(‡)] |
| • ViT-B/16 (☎) | 224 | 300 | 84.59 | 88.93 | 94.16 | 861 | 0.18k[(‡)] |
| • Mixer-L/16 | 224 | 300 | 82.89 | 87.54 | 93.63 | 419 | 0.41k[(‡)] |
| • ViT-L/16 (☎) | 224 | 300 | 84.46 | 88.35 | 94.49 | 280 | 0.55k[(‡)] |
| • Mixer-L/16 | 448 | 300 | 83.91 | 87.75 | 93.86 | 105 | 0.41k[(‡)] |
| Pre-trained on JFT-300M | | | | | | | |
| • Mixer-S/32 | 224 | 5 | 68.70 | 75.83 | 87.13 | 11489 | 0.01k |
| • Mixer-B/32 | 224 | 7 | 75.53 | 81.94 | 90.99 | 4208 | 0.05k |
| • Mixer-S/16 | 224 | 5 | 73.83 | 80.60 | 89.50 | 3994 | 0.03k |
| • BiT-R50x1 | 224 | 7 | 73.69 | 81.92 | — | 2159 | 0.08k |
| • Mixer-B/16 | 224 | 7 | 80.00 | 85.56 | 92.60 | 1384 | 0.08k |
| • Mixer-L/32 | 224 | 7 | 80.67 | 85.62 | 93.24 | 1314 | 0.12k |
| • BiT-R152x1 | 224 | 7 | 79.12 | 86.12 | — | 932 | 0.14k |
| • BiT-R50x2 | 224 | 7 | 78.92 | 86.06 | — | 890 | 0.14k |
| • BiT-R152x2 | 224 | 14 | 83.34 | 88.90 | — | 356 | 0.58k |
| • Mixer-L/16 | 224 | 7 | 84.05 | 88.14 | 94.51 | 419 | 0.23k |
| • Mixer-L/16 | 224 | 14 | 84.82 | 88.48 | 94.77 | 419 | 0.45k |
| • ViT-L/16 | 224 | 14 | 85.63 | 89.16 | 95.21 | 280 | 0.65k |
| • Mixer-H/14 | 224 | 14 | 86.32 | 89.14 | 95.49 | 194 | 1.01k |
| • BiT-R200x3 | 224 | 14 | 84.73 | 89.58 | — | 141 | 1.78k |
| • Mixer-L/16 | 448 | 14 | 86.78 | 89.72 | 95.13 | 105 | 0.45k |
| • ViT-H/14 | 224 | 14 | 86.65 | 89.56 | 95.57 | 87 | 2.30k |
| • ViT-L/16 [14] | 512 | 14 | 87.76 | 90.54 | 95.63 | 32 | 0.65k |

While not strictly comparable, this allows us to further extrapolate the effect of scale. We use the linear 5-shot top-1 accuracy on ImageNet as a proxy for transfer quality. For every pre-training run we perform early stopping based on the best upstream validation performance. Results are reported in Figure 2 (right), where we also include ViT-B/32, ViT-L/32, ViT-L/16, and BiT-R152x2 models.

When pre-trained on the smallest subset of JFT-300M, all Mixer models strongly overfit. BiT models also overfit, but to a lesser extent, possibly due to the strong inductive biases associated with the convolutions. As the dataset increases, the performance of both Mixer-L/32 and Mixer-L/16 grows faster than BiT; Mixer-L/16 keeps improving, while the BiT model plateaus.

The same conclusions hold for ViT, consistent with Dosovitskiy et al. [14]. However, the relative improvement of larger Mixer models are even more pronounced. The performance gap between Mixer-L/16 and ViT-L/16 shrinks with data scale. It appears that Mixer benefits from the growing dataset size even more than ViT. One could speculate and explain it again with the difference in inductive biases: self-attention layers in ViT lead to certain properties of the learned functions that are *less compatible* with the true underlying distribution than those discovered with Mixer architecture.

## 3.4 Invariance to input permutations

In this section, we study the difference between inductive biases of Mixer and CNN architectures. Specifically, we train Mixer-B/16 and ResNet50x1 models on JFT-300M following the pre-training

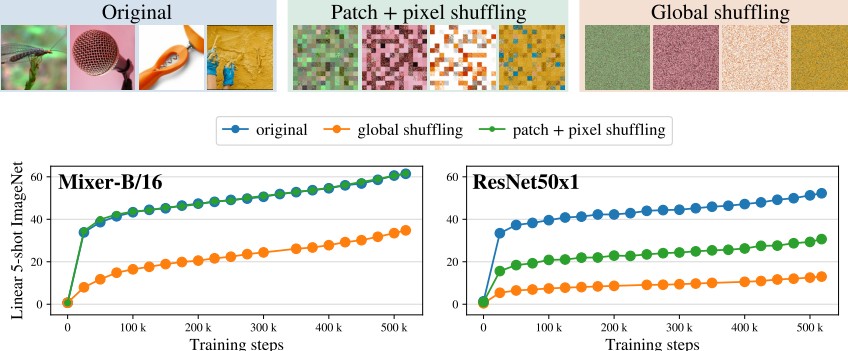

Figure 4: **Top:** Input examples from ImageNet before permuting the contents (left); after shuffling the $16 \times 16$ patches and pixels within the patches (center); after shuffling pixels globally (right). **Bottom:** Mixer-B/16 (left) and ResNet50x1 (right) trained with three corresponding input pipelines.

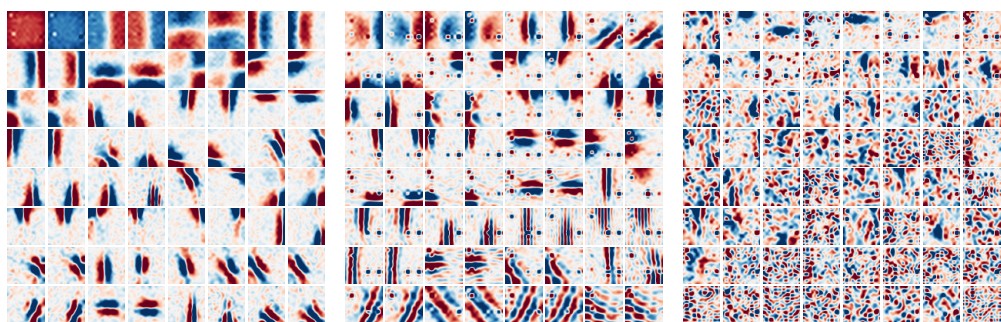

Figure 5: Hidden units in the first (**left**), second (**center**), and third (**right**) token-mixing MLPs of a Mixer-B/16 model trained on JFT-300M. Each unit has 196 weights, one for each of the $14 \times 14$ incoming patches. We pair the units to highlight the emergence of kernels of opposing phase. Pairs are sorted by filter frequency. In contrast to the kernels of convolutional filters, where each weight corresponds to one pixel in the input image, one weight in any plot from the left column corresponds to a particular $16 \times 16$ patch of the input image. Complete plots in Supplementary D.

setup described in Section 3 and using one of two different input transformations: (1) Shuffle the order of $16 \times 16$ patches and permute pixels within each patch with a shared permutation; (2) Permute the pixels globally in the entire image. Same permutation is used across all images. We report the linear 5-shot top-1 accuracy of the trained models on ImageNet in Figure 4 (bottom). Some original images along with their two transformed versions appear in Figure 4 (top). As could be expected, Mixer is invariant to the order of patches and pixels within the patches (the blue and green curves match perfectly). On the other hand, ResNet's strong inductive bias relies on a particular order of pixels within an image and its performance drops significantly when the patches are permuted. Remarkably, when globally permuting the pixels, Mixer's performance drops much less (~45% drop) compared to the ResNet (~75% drop).

### 3.5 Visualization

It is commonly observed that the first layers of CNNs tend to learn Gabor-like detectors that act on pixels in local regions of the image. In contrast, Mixer allows for global information exchange in the token-mixing MLPs, which begs the question whether it processes information in a similar fashion. Figure 5 shows hidden units of the first three token-mixing MLPs of Mixer trained on JFT-300M. Recall that the token-mixing MLPs allow global communication between different spatial locations. Some of the learned features operate on the entire image, while others operate on smaller regions. Deeper layers appear to have no clearly identifiable structure. Similar to CNNs, we observe many pairs of feature detectors with opposite phases [39]. The structure of learned units depends on the hyperparameters. Plots for the first embedding layer appear in Figure 2 of Supplementary D.

# 4 Related work

MLP-Mixer is a new architecture for computer vision that differs from previous successful architectures because it uses neither convolutional nor self-attention layers. Nevertheless, the design choices can be traced back to ideas from the literature on CNNs [24, 25] and Transformers [50].

CNNs have been the de-facto standard in computer vision since the AlexNet model [24] surpassed prevailing approaches based on hand-crafted image features [36]. Many works focused on improving the design of CNNs. Simonyan and Zisserman [41] demonstrated that one can train state-of-the-art models using only convolutions with small 3×3 kernels. He et al. [15] introduced skip-connections together with the batch normalization [20], which enabled training of very deep neural networks and further improved performance. A prominent line of research has investigated the benefits of using sparse convolutions, such as grouped [57] or depth-wise [9, 17] variants. In a similar spirit to our token-mixing MLPs, Wu et al. [55] share parameters in the depth-wise convolutions for natural language processing. Hu et al. [18] and Wang et al. [53] propose to augment convolutional networks with non-local operations to partially alleviate the constraint of local processing from CNNs. Mixer takes the idea of using convolutions with small kernels to the extreme: by reducing the kernel size to 1×1 it turns convolutions into standard dense matrix multiplications applied independently to each spatial location (channel-mixing MLPs). This alone does not allow aggregation of spatial information and to compensate we apply dense matrix multiplications that are applied to every feature across all spatial locations (token-mixing MLPs). In Mixer, matrix multiplications are applied row-wise or column-wise on the "patches×features" input table, which is also closely related to the work on sparse convolutions. Mixer uses skip-connections [15] and normalization layers [2, 20].

In computer vision, self-attention based Transformer architectures were initially applied for generative modeling [8, 34]. Their value for image recognition was demonstrated later, albeit in combination with a convolution-like locality bias [37], or on low-resolution images [10]. Dosovitskiy et al. [14] introduced ViT, a pure transformer model that has fewer locality biases, but scales well to large data. ViT achieves state-of-the-art performance on popular vision benchmarks while retaining the robustness of CNNs [6]. Touvron et al. [49] trained ViT effectively on smaller datasets using extensive regularization. Mixer borrows design choices from recent transformer-based architectures. The design of Mixer's MLP-blocks originates in [27, 50]. Converting images to a sequence of patches and directly processing embeddings of these patches originates in Dosovitskiy et al. [14].

Many recent works strive to design more effective architectures for vision. Srinivas et al. [42] replace 3×3 convolutions in ResNets by self-attention layers. Ramachandran et al. [37], Tay et al. [47], Li et al. [26], and Bello [3] design networks with new attention-like mechanisms. Mixer can be seen as a step in an orthogonal direction, without reliance on locality bias and attention mechanisms.

The work of Lin et al. [28] is closely related. It attains reasonable performance on CIFAR-10 using fully connected networks, heavy data augmentation, and pre-training with an auto-encoder. Neyshabur [31] devises custom regularization and optimization algorithms and trains a fully-connected network, attaining impressive performance on small-scale tasks. Instead we rely on token and channel-mixing MLPs, use standard regularization and optimization techniques, and scale to large data effectively.

Traditionally, networks evaluated on ImageNet [13] are trained from random initialization using Inception-style pre-processing [46]. For smaller datasets, transfer of ImageNet models is popular. However, modern state-of-the-art models typically use either weights pre-trained on larger datasets, or more recent data-augmentation and training strategies. For example, Dosovitskiy et al. [14], Kolesnikov et al. [22], Mahajan et al. [30], Pham et al. [35], Xie et al. [56] all advance state-of-the-art in image classification using large-scale pre-training. Examples of improvements due to augmentation or regularization changes include Cubuk et al. [11], who attain excellent classification performance with learned data augmentation, and Bello et al. [4], who show that canonical ResNets are still near state-of-the-art, if one uses recent training and augmentation strategies.

# 5 Conclusions

We describe a very simple architecture for vision. Our experiments demonstrate that it is as good as existing state-of-the-art methods in terms of the trade-off between accuracy and computational resources required for training and inference. We believe these results open many questions. On the practical side, it may be useful to study the features learned by the model and identify the main

differences (if any) from those learned by CNNs and Transformers. On the theoretical side, we would like to understand the inductive biases hidden in these various features and eventually their role in generalization. Most of all, we hope that our results spark further research, beyond the realms of established models based on convolutions and self-attention. It would be particularly interesting to see whether such a design works in NLP or other domains.

## Acknowledgments and Disclosure of Funding

The work was performed in the Brain teams in Berlin and Zürich. We thank Josip Djolonga for feedback on the initial version of the paper; Preetum Nakkiran for proposing to train MLP-Mixer on input images with shuffled pixels; Olivier Bousquet, Yann Dauphin, and Dirk Weissenborn for useful discussions.

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
