# Supplementary Materials for
# MLP-Mixer: An all-MLP Architecture for Vision

**Ilya Tolstikhin**[*]**, Neil Houlsby**[*]**, Alexander Kolesnikov**[*]**, Lucas Beyer**[*]**,**

**Xiaohua Zhai, Thomas Unterthiner, Jessica Yung, Andreas Steiner,**

**Daniel Keysers, Jakob Uszkoreit, Mario Lucic, Alexey Dosovitskiy**

[*]equal contribution

Google Research, Brain Team

```
{tolstikhin, neilhoulsby, akolesnikov, lbeyer,
  xzhai, unterthiner, jessicayung†, andstein,
keysers, usz, lucic, adosovitskiy}@google.com
```

[†]work done during Google AI Residency

## A    Things that did not help

### A.1    Modifying the token-mixing MLPs

We ablated a number of ideas trying to improve the token-mixing MLPs for Mixer models of various scales pre-trained on JFT-300M.

**Untying (not sharing) the parameters**    Token-mixing MLPs in the Mixer layer are shared across the columns of the input table $\mathbf{X} \in \mathbb{R}^{S \times C}$. In other words, the *same* MLP is applied to each of the $C$ different features. Instead, we could introduce $C$ *separate* MLPs with independent weights, effectively multiplying the number of parameters by $C$. We did not observe any noticeable improvements.

**Grouping the channels together**    Token-mixing MLPs take $S$-dimensional vectors as inputs. Every such vector contains values of a single feature across $S$ different spatial locations. In other words, token-mixing MLPs operate by looking at only *one channel* at once. One could instead group channels together by concatenating $G$ neighbouring columns in $\mathbf{X} \in \mathbb{R}^{S \times C}$, reshaping it to a matrix of dimension $(S \cdot G) \times (C/G)$. This increases the MLP's input dimensionality from $S$ to $G \cdot S$ and reduces the number of vectors to be processed from $C$ to $C/G$. Now the MLPs look at *several channels at once* when mixing the tokens. This concatenation of the column-vectors improved linear 5-shot top-1 accuracy on ImageNet by less than 1–2%.

We tried a different version, where we replace the simple reshaping described above with the following: (1) Introduce $G$ linear functions (with trainable parameters) projecting $\mathbb{R}^C$ to $\mathbb{R}^{C/G}$. (2) Using them, map each of the $S$ rows (tokens) in $\mathbf{X} \in \mathbb{R}^{S \times C}$ to $G$ different $(C/G)$-dimensional vectors. This results in $G$ different "views" on every token, each one consisting of $C/G$ features. (3) Finally, concatenate vectors corresponding to $G$ different views for each of the $C/G$ features. This results in a matrix of dimension $(S \cdot G) \times (C/G)$. The idea is that MLPs can look at $G$ different *views* of the original channels, when mixing the tokens. This version improved the top-5 ImageNet accuracy by 3–4% for the Mixer-S/32 architecture, however did not show any improvements for the larger scales.

**Pyramids**    All layers in Mixer retain the same, isotropic design. Recent improvements on the ViT architecture hint that this might not be ideal [9]. We tried using the token-mixing MLP to reduce the

Table 1: Hyperparameter settings used for pre-training Mixer models.

| Model | Dataset | Epochs | $lr$ | $wd$ | RandAug. | Mixup | Dropout | Stoch. depth |
|-------|---------|--------|------|------|----------|-------|---------|--------------|
| Mixer-B | ImNet | 300 | 0.001 | 0.1 | 15 | 0.5 | 0.0 | 0.1 |
| Mixer-L | ImNet | 300 | 0.001 | 0.1 | 15 | 0.5 | 0.0 | 0.1 |
| Mixer-B | ImNet-21k | 300 | 0.001 | 0.1 | 10 | 0.2 | 0.0 | 0.1 |
| Mixer-L | ImNet-21k | 300 | 0.001 | 0.1 | 20 | 0.5 | 0.0 | 0.1 |
| Mixer-S | JFT-300M | 5 | 0.003 | 0.03 | – | – | – | – |
| Mixer-B | JFT-300M | 7 | 0.003 | 0.03 | – | – | – | – |
| Mixer-L | JFT-300M | 7/14 | 0.001 | 0.03 | – | – | – | – |
| Mixer-H | JFT-300M | 14 | 0.001 | 0.03 | – | – | – | – |

number of tokens by mapping from $S$ input tokens to $S' < S$ output tokens. While first experiments showed that on JFT-300M such models significantly reduced training time without losing much performance, we were unable to transfer these findings to ImageNet or ImageNet-21k. However, since pyramids are a popular design, exploring this design for other vision tasks may still be promising.

### A.2   Fine-tuning

Following ideas from BiT [4] and ViT [2], we also tried using mixup [10] and Polyak averaging [5] during fine-tuning. However, these did not lead to consistent improvements, so we dropped them. We also experimented with using inception cropping [7] during fine-tuning, which also did not lead to any improvements. We did these experiments for JFT-300M pre-trained Mixer models of all scales.

## B   Pre-training: hyperparameters, data augmentation and regularization

In Table 1 we describe optimal hyperparameter settings that were used for pre-training Mixer models.

For pre-training on ImageNet and ImageNet-21k we used additional augmentation and regularization. For RandAugment [1] we always use two augmentations layers and sweep magnitude, $m$, parameter in a set $\{0, 10, 15, 20\}$. For mixup [10] we sweep mixing strength, $p$, in a set $\{0.0, 0.2, 0.5, 0.8\}$. For dropout [6] we try dropping rates, $d$ of 0.0 and 0.1. For stochastic depth, following the original paper [3], we linearly increase the probability of dropping a layer from 0.0 (for the first MLP) to $s$ (for the last MLP), where we try $s \in \{0.0, 0.1\}$. Finally, we sweep learning rate, $lr$, and weight decay, $wd$, from $\{0.003, 0.001\}$ and $\{0.1, 0.01\}$ respectively.

## C   Fine-tuning: hyperparameters and higher image resolution

Models are fine-tuned at resolution 224 unless mentioned otherwise. We follow the setup of [2]. The only differences are: (1) We exclude $lr = 0.001$ from the grid search and instead include $lr = 0.06$ for CIFAR-10, CIFAR-100, Flowers, and Pets. (2) We perform a grid search over $lr \in \{0.003, 0.01, 0.03\}$ for VTAB-1k. (3) We try two different ways of pre-processing during evaluation: (i) "resize-crop": first resize the image to $256 \times 256$ pixels and then take a $224 \times 224$ pixel sized central crop. (ii) "resmall-crop": first resize the shorter side of the image to 256 pixels and then take a $224 \times 224$ pixel sized central crop. For the Mixer and ViT models reported in Table 3 of the main text we used (ii) on ImageNet, Pets, Flowers, CIFAR-10 and CIFAR-100. We used the same setup for the BiT models reported in Table 3 of the main text, with the only exception of using (i) on ImageNet. For the Mixer models reported in Table 2 of the main text we used (i) for all 5 downstream datasets.

Fine-tuning at higher resolution than the one used at pre-training time has been shown to substantially improve the transfer performance of existing vision models [8, 4, 2]. We therefore apply this technique to Mixer as well. When feeding images of higher resolution to the model, we do not change the patch size, which results in a longer sequence of tokens. The token-mixing MLPs have to be adjusted to handle these longer sequences. We experimented with several options and describe the most successful one below.

Table 2: Further details on computational complexity for the models in Table 3. Throughputs are measured in images/sec/core.

| | Image size | Pre-Train Epochs | Pre-Train exaFLOPs | Throughput TPUv3 | Throughput GPU V100 |
|---|---|---|---|---|---|
| ● Mixer-S/32 | 224 | 5 | 9 | 11489 | 5497 |
| ● Mixer-B/32 | 224 | 7 | 41 | 4208 | 1845 |
| ● Mixer-S/16 | 224 | 5 | 34 | 3994 | 1605 |
| ● BiT-R50x1 | 224 | 7 | 50 | 2159 | 1553 |
| ● Mixer-B/16 | 224 | 7 | 161 | 1384 | 516 |
| ● Mixer-L/32 | 224 | 7 | 145 | 1314 | 546 |
| ● BiT-R152x1 | 224 | 7 | 141 | 932 | 639 |
| ● BiT-R50x2 | 224 | 7 | 199 | 890 | 481 |
| ● BiT-R152x2 | 224 | 14 | 1126 | 356 | 192 |
| ● Mixer-L/16 | 224 | 14 | 1141 | 419 | 151 |
| ● ViT-L/16 | 224 | 14 | 1567 | 280 | 100 |
| ● Mixer-H/14 | 224 | 14 | 3096 | 194 | 58 |
| ● BiT-R200x3 | 224 | 14 | 3306 | 141 | 71 |
| ● ViT-H/14 | 224 | 14 | 4262 | 87 | 37 |

For simplicity we assume that the image resolution is increased by an integer factor $K$. The length $S$ of the token sequence increases by a factor of $K^2$. We increase the hidden width $D_S$ of the token-mixing MLP by a factor of $K^2$ as well. Now we need to initialize the parameters of this new (larger) MLP with the parameters of the pre-trained MLP. To this end we split the input sequence into $K^2$ equal parts, each one of the original length $S$, and initialize the new MLP so that it processes all these parts independently in parallel with the pre-trained MLP.

Formally, the pre-trained weight matrix $\mathbf{W}_1 \in \mathbb{R}^{D_S \times S}$ of the original MLP in Eq. 1 of the main text will be now replaced with a larger matrix $\mathbf{W}'_1 \in \mathbb{R}^{(K^2 \cdot D_S) \times (K^2 \cdot S)}$. Assume the token sequence for the resized input image is a concatenation of $K^2$ token sequences of length $S$ each, computed by splitting the input into $K \times K$ equal parts spatially. We then initialize $\mathbf{W}'_1$ with a block-diagonal matrix that has copies of $\mathbf{W}_1$ on its main diagonal. Other parameters of the MLP are handled analogously.

## D   Weight visualizations

For better visualization, we sort all hidden units according to a heuristic that tries to show low frequency filters first. For each unit, we also try to identify the unit that is closest to its inverse. Figure 1 shows each unit followed by its closest inverse. Note that the models pre-trained on ImageNet and ImageNet-21k used heavy data augmentation. We found that this strongly influences the structure of the learned units.

We also visualize the linear projection units in the embedding layer learned by different models in Figure 2. Interestingly, it appears that their properties strongly depend on the patch resolution used by the models. Across all Mixer model scales, using patches of higher resolution 32×32 leads to Gabor-like low-frequency linear projection units, while for the 16×16 resolution the units show no such structure.

## E   More details on computational cost

In Table 2 we report additional information on the computational cost of various models considered in this paper, including test-time throughput on V100 GPUs and number of FLOPs during pre-training.

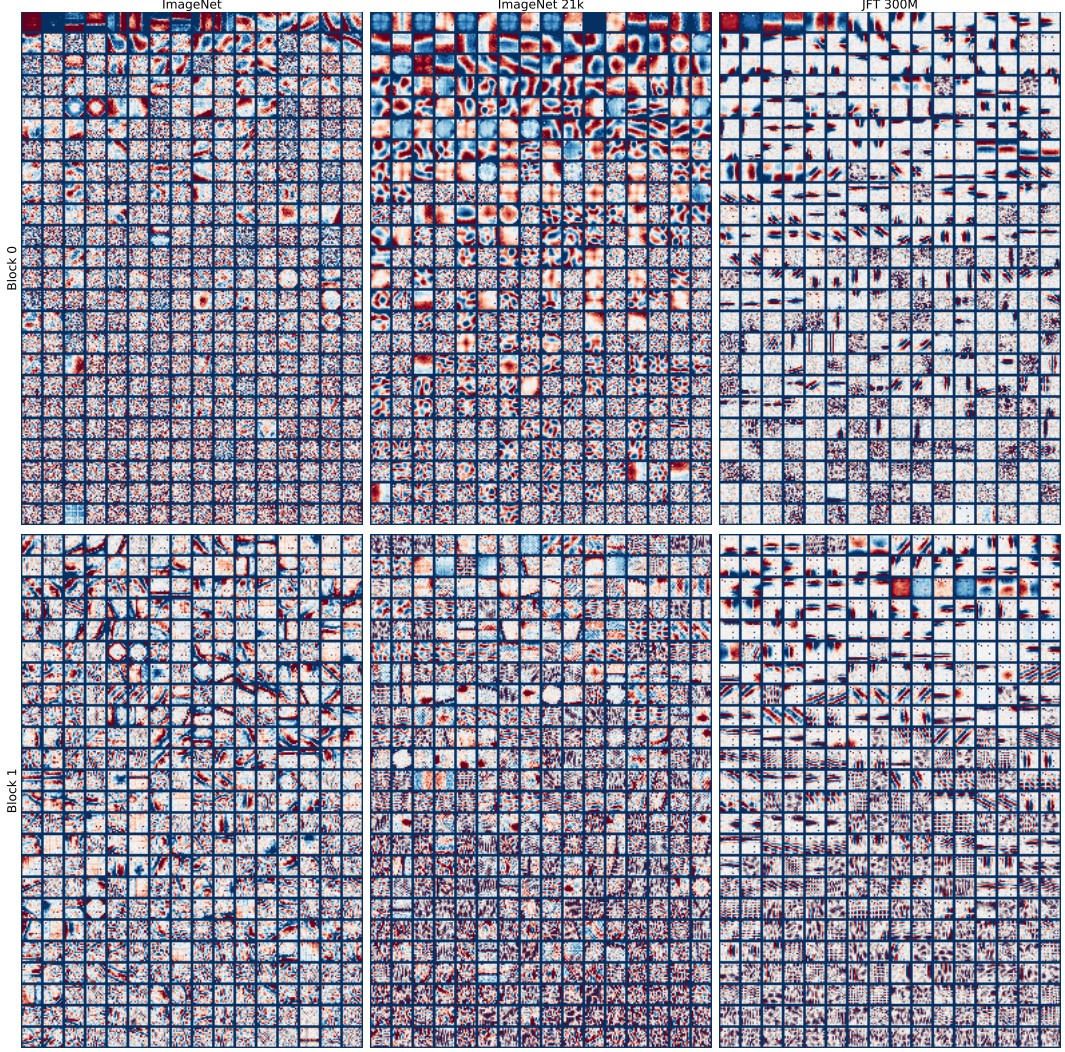

Figure 1: Weights of all hidden dense units in the first two token-mixing MLPs (**rows**) of the Mixer-B/16 model trained on three different datasets (**columns**). Each unit has $14 \times 14 = 196$ weights, which is the number of incoming tokens, and is depicted as a $14 \times 14$ image. In each block there are 384 hidden units in total.

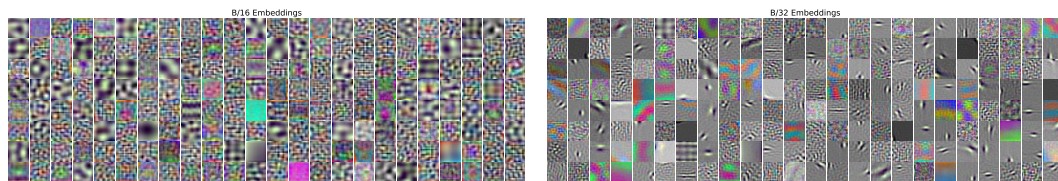

Figure 2: Linear projection units of the embedding layer for Mixer-B/16 (**left**) and Mixer-B/32 (**right**) models pre-trained on JFT-300M. Mixer-B/32 model that uses patches of higher resolution $32 \times 32$ learns very structured low frequency projection units, while most of the units learned by the Mixer-B/16 have high frequencies and no clear structure.

# F MLP-Mixer code

```python
import einops
import flax.linen as nn
import jax.numpy as jnp

class MlpBlock(nn.Module):
  mlp_dim: int
  @nn.compact
  def __call__(self, x):
    y = nn.Dense(self.mlp_dim)(x)
    y = nn.gelu(y)
    return nn.Dense(x.shape[-1])(y)

class MixerBlock(nn.Module):
  tokens_mlp_dim: int
  channels_mlp_dim: int
  @nn.compact
  def __call__(self, x):
    y = nn.LayerNorm()(x)
    y = jnp.swapaxes(y, 1, 2)
    y = MlpBlock(self.tokens_mlp_dim, name='token_mixing')(y)
    y = jnp.swapaxes(y, 1, 2)
    x = x+y
    y = nn.LayerNorm()(x)
    return x+MlpBlock(self.channels_mlp_dim, name='channel_mixing')(y)

class MlpMixer(nn.Module):
  num_classes: int
  num_blocks: int
  patch_size: int
  hidden_dim: int
  tokens_mlp_dim: int
  channels_mlp_dim: int
  @nn.compact
  def __call__(self, x):
    s = self.patch_size
    x = nn.Conv(self.hidden_dim, (s,s), strides=(s,s), name='stem')(x)
    x = einops.rearrange(x, 'n h w c -> n (h w) c')
    for _ in range(self.num_blocks):
      x = MixerBlock(self.tokens_mlp_dim, self.channels_mlp_dim)(x)
    x = nn.LayerNorm(name='pre_head_layer_norm')(x)
    x = jnp.mean(x, axis=1)
    return nn.Dense(self.num_classes, name='head',
                    kernel_init=nn.initializers.zeros)(x)
```

Listing 1: MLP-Mixer code written in JAX/Flax.