# OpenReview forum: "MLP-Mixer: An all-MLP Architecture for Vision"
_NeurIPS.cc/2021/Conference — NeurIPS 2021 Poster_

### Official Review · Reviewer_tsff · 2021-07-13

**Rating:** 6
**Confidence:** 5

**Summary:**

The paper proposes to use MLPs in Image Classification and obtains correct performances on ImageNet and VTAB .
Their architecture is similar to vision transformers except that it replaces the self attention with an MLP (similar to the FFN of transformers) applied on the tokens.

**Limitations And Societal Impact:**

Yes the authors adequately addressed the limitations and potential negative societal impact of their work

**Main Review:**


Strengths:
- The results obtained are good for an MLP architecture that is simple. This results are very interesting and very aligned with the recent work on vision transformers in computer vision.
- Results are reported on ImageNet, ImageNet-real and VTAB  with pre-training with different amount of date which is a good practice and very useful. Nevertheless it would be interesting to have also the results on ImageNet-v2 [1].
- The paper is well written and easy to follow.


Weakness:
- The comparison on ImageNet without extra-training data with other types of architecture is quite incomplete. There are only comparisons with BiT and ViT which are not state of the art in Image classification for convnets and transformers architecture.
It would be interesting to have comparisons with EfficientNet-v2 [2], NFNet [3], CaiT [4], Swin [5] or CvT [6].
For the comparison on ImageNet with extra-data, MLP-mixer is compared with more architecture but it lacks some state-of-the-art approaches like EfficientNet-v2 [2], Swin [5] or CvT[6].
This may bias the conclusion: "MLP-Mixer attains competitive scores on image classification benchmarks, with pre-training and inference cost comparable to state-of-the-art models" L9
Nevertheless, the paper indicates that the MLP mixer is still below ViT: "When trained from scratch on ImageNet, Mixer-B/16 achieves a reasonable top-1 accuracy of 76.44%. This is 3% behind the ViT-B/16 model" L182

Minor comments:

- The ViT results pre-trained on ImageNet with extra regularization are quite low in comparison to the DeiT paper. One may wonder why the paper does not use the same training procedure.

- FLOPs of the models are not reported. It would be very useful for the comparison with other architecture of the literature since it is a fairly commonly used metric. This metric is not perfect but the throughput also has some flaws as it depends a lot on the implementation, the hardware, ....

[1] Recht et al.,  Do ImageNet Classifiers Generalize to ImageNet?
[2] Tan et al., EfficientNetV2: Smaller Models and Faster Training
[3] Brock et al. High-Performance Large-Scale Image Recognition Without Normalization
[4] Touvron et al. Going deeper with Image Transformers
[5] Liu et al. Swin Transformer: Hierarchical Vision Transformer using Shifted Windows
[6] Wu et al. CvT: Introducing Convolutions to Vision Transformers


**Time Spent Reviewing:**

3

---

> ### Author Response · Authors · 2021-08-10
> **Response to Reviewer tsff**
>
> We address your questions one by one.
>
> * 1. Results on ImageNet without extra-data pre-training.
>
> Since MLP-Mixer benefits from larger-scale training, we focus on absolute ImageNet performance, and only provide ImageNet from-scratch performance as an ablation test. We will highlight that it is a non-goal for this paper to be competitive at from-scratch training, and that models exclusively designed for ImageNet from-scratch perform significantly better than MLP-Mixer; note, this is already mentioned in line 161 for BottleNeck Transformer and NFNet.
>
> * 2. Results on ImageNet with extra-data pre-training. Comparisons to EfficientNetv2, Swin, CvT lacking.
>
> We do not claim state-of-the-art performance with pre-training. We claim that the proposed novel and extremely simple architecture achieves results comparable (surprisingly) to the other popular models.
>
> Concurrent work, such as EfficientNet-v2 [2], Swin [5], and CvT [6] (as well as CaiT [4]) all appeared on arXiv between end of March and beginning of April, 2021, i.e. ~1 month before the submission deadline. It would not have been possible to perform proper comparisons in that time period. However, the numbers reported there (EfficientNet-v2 gets 87.3, Swin 86.4, CvT 87.7) are similar to the numbers in Table 2 and we believe do not change the conclusions made in the paper.
>
> * 3. From scratch results of ViT on ImageNet are lower than DeiT numbers.
>
> We compare MLP-Mixer to ViT models in a controlled setting using a carefully designed unified regularization and augmentation sweep covering a wide range of settings from [Steiner et al., 2021]. This allows us to draw robust conclusions about relative performances of these models. Using the numbers reported in [Touvron et al., 2020] paper would not be an apples-to-apples comparison.
>
> [Touvron et al., 2020] H. Touvron, M. Cord, M. Douze, F. Massa, A. Sablayrolles, and H. Jégou. Training data-efficient image transformers & distillation through attention. 2020.
>
> [Steiner et al., 2021] A. Steiner, A. Kolesnikov, X. Zhai, R. Wightman, J. Uszkoreit, L. Beyer. “How to train your ViT? Data, Augmentation, and Regularization in Vision Transformers”. 2021
>
> * 4. FLOPS not reported.
>
> We will include FLOPs for Mixer and ViT models in the updated version of the text (together with throughputs on GPU).

---

> > ### Comment · Reviewer_tsff · 2021-09-03
> > **Thanks for your response**
> >
> > Thank you for your response, the paper proposes an interesting idea however the rebuttal partially answer my concerns:
> >
> > - The answer 2) does not satisfy me because it is easy and fast to report the results presented in previous papers.
> > - The answer 3) does not satisfy me either. A lot of papers on vision transformers have adopted the DeiT training procedure. Using it would have allowed a better comparison with other architectures. It is difficult to know if the ViT baseline is in this context tuned as well as the MLP mixer architecture.
> >
> > So I slightly lower my initial rating (7 -> 6).

---

### Official Review · Reviewer_6miq · 2021-07-15

**Rating:** 6
**Confidence:** 5

**Summary:**

This paper proposes a neural network architecture of all MLP layers without any convolution or attention layers. Although it only has MLP layers, it still needs to exchange the information across spacial locations by token mixing strategy. By pretraining on large-scale private datasets, the proposed method achieve comparable results as the standard CNN and transformer networks.

**Limitations And Societal Impact:**

No much about limitations. See above.

**Main Review:**

-This paper is the first work show strong results by only using MLP layers. But this idea is not too difficult to think, given the recent great successes of transformer based vision (ViT) models. Most of techniques and experiences are from ViT, and the only changes is to replace the attention layers by the MLP layers.

-The quality of the papers is good. Experiments are sufficient and most of the claims are supported. But this paper is a little bit obsessed with classification task, especially on ImageNet. I don't quite understand the point to push forward the ImageNet top-1 accuracy, e.g. over 90%, by pretraining a super large model on a super large private dataset, especially when ImageNet itself has some annotation errors. And Table 3 reports "Avg.5"? VTAB is quite common for this kind of Google papers.

-The paper is quite clear and easy to read.

-I am not confident about the significance of this paper. We can say it is a reduced alternative to ViT, by replacing attention layer with token-mixing MLP layer. But we already have CNN and ViT, why do we still need MLP-Mixer? As claimed in the paper, it is an alternative to CNN and ViT, then what are the advantages and disadvantages of MLP-Mixer, for classification and beyond classification? I don't think MLP-Mixer can replace CNN or transformer and can do some tasks that are unavailable for CNN or transformer.

-I think this paper overclaim the significance of MLP-Mixer as alternative to CNN or transformer. As mentioned in the paper, it is a special case of CNN, with 1x1 and global depth-wise convolution. How could a special case be an alternative for a general case?

================after rebuttal================

I am disappointed by the rebuttal. The rebuttal doesn't have satisfactory responses to the two major concerns: 1) the application limit of this paper and 2) the overclaimed significance. What's worse is that the authors seem not to be willing to improve them. Given these, I will downgrade my rating.

**Time Spent Reviewing:**

3

---

> ### Author Response · Authors · 2021-08-10
> **Response to Reviewer 6miq**
>
> We address your questions one by one.
>
> * 1. Significance of the results.
>
> We address this question in the general comment.
>
> * 2. What is the point of pushing forward the ImageNet top-1 accuracy further?
>
> It has been shown that ImageNet performance (even at high scores) is an excellent proxy for general-purpose performance in Computer Vision. See [Kornblith et al., 2018] who show extensively that better ImageNet models transfer better to many other tasks. We also confirm this by showing transfer results to over 20 datasets (VTAB). That being said, we agree that there are other important tasks in computer vision, beyond classification. In this paper we focus on classification and demonstrate that for this central task a network built up of MLPs can work well. Extending to other tasks is an open area for future research.
>
> [Kornblith et al., 2018] Simon Kornblith, Jonathon Shlens, Quoc V. Le. “Do Better ImageNet Models Transfer Better?”
>
> * 3. MLP-Mixer is a special case of CNNs. How could a special case be an alternative for a general case?
>
> The token-mixing MLPs consist of two fully connected layers with a non-linearity sitting in-between. These can not be implemented using a single (or even two) global depthwise convolutions. In the paper we say that the way in which MLP-Mixer implements the between-patch communication is *similar* to using a single-channel depth-wise global convolutions with parameter sharing, but the two are not equivalent.
>
> * 4. “Avg.5” and VTAB.
>
> VTAB results are available in Table 2. “Avg. 5” stands for the average performance across five downstream tasks: ImageNet, CIFAR-10, CIFAR-100, Pets, Flowers (see beginning of Section 3.1).

---

### Official Review · Reviewer_MVp5 · 2021-07-18

**Rating:** 6
**Confidence:** 5

**Summary:**

This paper presents an all-MLP based architecture for visual recognition tasks. The presented model is to replace the multi-head self-attention block with a spatial MLP in the original Transformer model. With a large-scale pre-trained dataset, this model with few inductive biases could achieve relatively good performance on image classification tasks.

**Ethics Review Area:**

["Privacy and Security (e.g., consent)"]

**Limitations And Societal Impact:**

Yes.

**Main Review:**

For originality, the architecture proposed in this paper is simple and somewhat novel. A spatial MLP is an effective approach to perform spatial connections when a large-scale dataset is available.

The paper is well written and well organized.

With a large-scale pre-trained dataset, this model with few inductive biases could achieve relatively good performance on image classification tasks. This observation is interesting and valuable to the community.

My major concern is about the spatial MLP (MLP1). This is a simple design, but the application scenario is also limited due to the fixed size of the input spatial resolution. For example, for some downstream tasks of object detection and semantic segmentation, the input sizes are different to that of the image classification. This architecture could not be directly applied to these tasks.

Also, it is interesting to see the performance of the proposed approach on natural language processing tasks, such as in comparison to BERT on mask language modeling.

**Needs Ethics Review:**

Yes

**Time Spent Reviewing:**

5

---

> ### Author Response · Authors · 2021-08-10
> **Response to Reviewer MVp5**
>
> We address your point on spatial MLPs (difficulties when using MLP-Mixer with various resolutions) in a separate general comment.
>
> Regarding the usefulness of the proposed approach on NLP tasks, this would be an interesting extension. In fact we became aware of several concurrent and follow-up works (which we cannot link here, to preserve anonymity) that apply similar techniques to the NLP tasks and report promising preliminary results.

---

### Official Review · Reviewer_5HxS · 2021-07-18

**Rating:** 5
**Confidence:** 5

**Summary:**

The paper proposes to a new architecture for image classification, based on recently proposed Vision Transformers. Instead of employing self-attention as a global operation on patch embeddings, the authors use fully connected layers, followed by channel mixing done with fully connected layers too. The changes are motivated by improving performance on proprietary hardware (TPU), designed for matrix multiplication. Unlike ViT which has quadratic cost on the number of input patches, the proposed Mixer model has linear cost. However, it loses the ability to handle arbitrary image resolutions. The experimental evaluation shows that the proposed model is comparable performance to state-of-the-art convolutional and Transformer-based models on proprietary and public data.

**Limitations And Societal Impact:**

The authors addressed the limitations and potential negative societal impact of their work adequately.

**Main Review:**

Originality: the idea of splitting image classification networks into mixing spatially and channel-wise is not novel and was explored in Network-in-Network [Lin et al, 2014], in which convolutions were interleaved with MLPs too. This important citation and comparison are missing in the paper. The authors motivate Mixer with "a convolution is more complex than plain matrix multiplication in MLPs as it requires an additional costly reduction to matrix multiplication  and/or specialized implementation" (lines 36-37). I disagree with this statement. The concepts of matrix multiplication and convolution are both equally simple, and matrix multiplication requires a specialized implementation as well. The motivation to use matrix multiplication comes from the usage of proprietary hardware (TPU), which was originally designed for matrix multiplication, and is not well suited to convolution. The paper thus addresses an issue that only exists within one company which has access to the particular proprietary hardware and large proprietary datasets. This needs to be clearly stated in motivation. Compared to Network-In-Network, the authors show that spatial mixing can be done with MLP instead of convolution, or self-attention as in ViT, without the need of positional encoding, which is an interesting and novel finding. However, such reduction of inductive bias in architecture comes at a price:
- the model cannot be applied to images of different resolutions easily (as the authors state in section 3), as unlike CNN or ViT it cannot be applied in "fully-convolutional" manner. There are at least two very important cases for this, finetuning and object detection;
- the model becomes more data-hungry;
- the number of model parameters should increase: that would be expected, and important for deployment of such networks, yet never discussed in the paper.
The paper lacks related work on extremely simple and high performant CNN architectures, e.g. RepVGG [Ding et al. 2021], consisting only of convolutions and nonlinearities.

Quality: The authors provide an extensive evaluation and test the model for invariance to input permutations. Tables miss the indication of numbers of parameters for different models. I would be valuable to include throughput on other devices than TPUs, e.g. GPUs.

Clarity: the paper is well written and is easy to follow.

Significance: the finding that image classification can be done with 1x1 convolutions interleaved with MLPs for spatial mixing is signficant, but will mostly be interesting for the users of the proprietary devices for which the model was designed for, as it also loses some desirable features of CNN and ViT.

**Time Spent Reviewing:**

4

---

> ### Author Response · Authors · 2021-08-10
> **Response to Reviewer 5HxS**
>
> We address your questions one by one.
>
> We note first that most of these concerns are of a practical nature (e.g. other recent networks might be more parameter efficient, how to extend MLP-Mixer to object detection, etc.). While we believe that MLP-Mixer is a highly practical design, in part, due to the massive benefit of an almost-trivial implementation, this misses one of the main points of the paper: a conceptually simple architecture consisting only of MLP blocks achieving results comparable to other mature Computer Vision models.
>
> * 1. Number of parameters not reported.
>
> We provide parameter numbers for all MLP-Mixer models in Table 1 (last row).
>
> * 2. Comparisons to Network-in-Network [Lin et al., 2014].
>
> Thank you for pointing out this work. Indeed, composing the DNN architecture using cross-patch and per-patch operations is not novel (and we did not claim otherwise). NIN implements (a) the per-patch operation in a very similar way to ViT’s “MLP Blocks” and MLP-Mixer’s “channel-mixing MLPs” and (b) the cross-patch operations by sliding the MLPs over the input with overlapping patches, similarly to CNNs. In contrast, MLP-Mixer performs (b) also using MLPs, as pointed out by the reviewer, which is the main novelty in the model. We will include this discussion to the updated version of the text.
>
> * 3. The motivation to use matrix multiplication comes from the usage of proprietary hardware (TPU). [...] issue that only exists within one company [...].
>
> We would like to respectfully disagree. While obviously not as widespread as CPUs and GPUs, TPUs are publicly available and will hopefully become more and more common in research thanks to various dedicated efforts (e.g. https://huggingface.co/blog/pytorch-xla). We believe that eventually, as new models that scale efficiently on TPUs are discovered and their pre-trained checkpoints are made public, the ML/DNN community as a whole will benefit.
>
> * 4. [TPU] was originally designed for matrix multiplication, and is not well suited to convolution.
>
> We made additional experiments measuring throughputs of ResNet50x1 and Mixer-B/16 models on GPUs (V100). Here are the results:
> ResNet50: 1553 (GPU) / 2159 (TPU). GPU->TPU speedup: ~1.4x.
> MLP-Mixer-B/16: 516 (GPU) / 1384 (TPU). GPU->TPU speedup: ~2.7x.
> It is clear that ResNets *do run faster* on TPUs than on GPUs. Indeed, the speedup is not as large as for the MLP-Mixer, but it is not true that “TPUs are not well suited for CNNs”.
> We will add the table with throughputs on GPU for various ResNet, MLP-Mixer, and ViT models in the appendix of the updated text.
>
> * 5. However, such reduction [token-mixing MLPs] of inductive bias in architecture comes at a price: the model [MLP-Mixer] becomes more data-hungry.
>
> We agree with the reviewer. Moreover, this claim is equally true for both Vision Transformer and MLP-Mixer: compared to CNNs, both models require more data (or more aggressive augmentation techniques) to achieve the same performance. Recent works [Touvron et al., 2020, Steiner et al., 2021] have shown that additional regularization and tricks can be used instead of data for Vision Transformers; we believe similar advances will be worth exploring for MLP-Mixer also.
>
> [Touvron et al., 2020] H. Touvron, M. Cord, M. Douze, F. Massa, A. Sablayrolles, and H. Jégou. Training data-efficient image transformers & distillation through attention. 2020.
>
> [Steiner et al., 2021] A. Steiner, A. Kolesnikov, X. Zhai, R. Wightman, J. Uszkoreit, L. Beyer. “How to train your ViT? Data, Augmentation, and Regularization in Vision Transformers”. 2021
>
> * 6. Comparisons to RepVGG.
>
> There are very many CNN/attention variants that we could compare to, including RepVGG. Since we focus on pre-training we compared to state-of-the-art networks that also use pre-training and attain scores significantly higher than reported in RepVGG.

---

### Review · Ethics_Reviewer_k7bd · 2021-08-11

**Recommendation:**

It is possible for the researchers to address the 2 ethical concerns raised above. They do not have to 'fix' them but they should be able to speak to how these concerns factor into their sense of the value of their proposed model.

**Ethical Issues:**

Yes

**Ethics Review:**

The paper offers an analysis of an image recognition architecture, MLP-Mixer, arguing that the approach, while data-hungry and dependent on proprietary data for training, can do as good a job as more compute-costly approaches, like CNNs.

The ethical issues raised in the reviews seem to be two-fold: 1) there's no sense of the provenance/foundational consent that would outline the rights of the researchers to train on the proprietary data used for this technique and (more importantly) 2) the researchers are proposing a technique that necessarily requires more real-world end-user data. Both of these components of the research should prompt researchers to address the role that rights to people's data do or do not play in their thinking about the value of this technique.

---

> ### Author Response · Authors · 2021-09-01
> **Response to Ethics Reviewer k7bd**
>
> Q: “What are the social implications of creating models even more dependent on data capture and retention for modeling?”
>
> Following [1] (referenced among others in the white paper [2]), the first question is “What are the possible applications outside of the lab”? We developed a DNN architecture for computer vision and shared the pre-trained checkpoints. In principle, applications could be very broad, as the architecture (as well as the checkpoints) could be used in many CV tasks (medical imaging, player pose tracking in sports, crop monitoring in agriculture, tracking astronomical objects in space science, etc). Likely some of these applications will affect our everyday life in the future.
>
> Next we list some of the hypothetical societal impacts of our results, both positive and negative. Following the reviewer’s suggestion, we will try to focus on such aspects as large scale of data, its capture and retention. In order to make the discussion more concrete in some cases we will use the hypothetical case of applying such vision models to a medical image analysis task.
>
> Negative effects:
> * Collecting large datasets is an expensive process (even if the process is semi-automatic). It will limit the ability of many academic labs and companies to participate in the research/applied efforts in case “data-hungry” models will become mainstream. The research community as a whole may alleviate this problem by creating joint efforts or by sharing access to created datasets, as has often happened in the past.
>
> * Larger datasets are likely harder to manually inspect for various biases. “Are all the doctors who collected the images male?” This and similar questions are easier to answer for smaller datasets (1 million images) than for larger ones (billions of images).
>
> * There may be important applied tasks (eg. involving rare astronomical events) where collecting large datasets is impossible. If “data-hungry” models become mainstream and the ML/CV community focuses on them exclusively, we may lack improved techniques designed for small datasets. In some cases, the smaller datasets can be solved with transfer learning using the models pre-trained with other larger datasets. However, sometimes transfer may fail (eg. if the new dataset is exceptional and very different from everything else). The community should keep working on both (a) creating models that perform well while using less training data and (b) creating models that perform as well as possible while using large amounts of data.
>
> Positive effects:
> * Calibration / uncertainty. Recently published research (which we cannot link here, to preserve anonymity) suggests that MLP-Mixer is well calibrated and that it is robust to distribution shift (on par with ViT). This may be due to both data scale and architectural features. Better calibration (i.e. better prediction of model’s uncertainty) could be instrumental in various mission-critical systems like medical imaging (e.g. a system aiding human doctors in cancer tumor detection), where false positives are dangerous. With well calibrated models the doctors could have a better idea of when their attention is required more. Robustness to distribution shift is another important property of the learning systems. Imagine a medical imaging system based on DNNs integrated in certain hardware. One day the hospital updates the sensor in the hardware, as part of the planned maintenance. It turns out that the sensor results in a slightly different resolution (or brightness, or contrast, etc) of the images. This could lead to a sudden (and perhaps even un-noticed) drop in model’s accuracy. Improved robustness could potentially help in this and similar situations.
>
> * Long tail (aka “rare events”) and its role in real world data is a common topic these days, well documented for instance [3] (for ImageNet). If the training set is large, it likely contains plenty (in absolute terms) of rare events from the “long tail”. This could be useful for the model to learn those rare events. Most likely, modeling them while using smaller datasets could lead to difficult questions of balancing the datasets (or relying on “memorization”, instead of “learning”). Note that proper modeling of rare events is often of a great importance, i.e. diagnosing rare diseases in medical imaging, facing rare weather conditions in autonomous driving, etc.
>
> * Think of applications where unlimited training datasets could be easily and “safely” obtained. In such cases there is really no problem (apart, perhaps, from computational costs) in using “data-hungry” models? After all, “increase the training set” is one of the most classical and well studied recipes for reducing the test error (VC and PAC theories of learning).
>
> [1] https://medium.com/@GovAI/a-guide-to-writing-the-neurips-impact-statement-4293b723f832
>
> [2] https://partnershiponai.org/paper/responsible-publication-recommendations/
>
> [3] https://arxiv.org/pdf/1906.05271.pdf

---

### Author Response · Authors · 2021-08-10
**General comments**

We thank the reviewers for their work and valuable feedback. We address all 4 reviews in separate replies. Here we would like to address two common questions.

* 1. The model can not be easily applied to various resolutions. Difficulty applying to tasks like detection and segmentation.

First, we demonstrate that pre-trained MLP-Mixer models can be successfully fine-tuned with various resolutions (e.g. main results in Table 2). However, we agree that this requires more effort than with convolutional models, e.g. ResNets.

We highlight to the reviewers that MLP-Mixer is not meant to be an immediate replacement for other popular models that were developed for several years, but instead to demonstrate the surprising performance from a new and very simple design. We hope that in the future the ideas presented in the paper will be useful in tasks beyond classification, including segmentation and detection.

* 2. Significance?

It is extremely surprising, from a research perspective, that a model consisting purely of MLPs can be competitive, even with SOTA CNNs and Attention-based models that have been developed for many years. This simple MLP-based architecture has already had a very significant influence on the community: there are several follow up papers already (which we cannot link here, to preserve anonymity). Therefore, we believe that Mixer opens a large research avenue for extensions and improvements to further increase usage by practitioners.

---

### Decision · Program_Chairs · 2021-09-27

**Decision:**

Accept (Poster)

**Comment:**

Initially, the paper received scores 7, 7, 6, and 5. All reviewers consider the idea is very interesting, experiments are comprehensive, the paper is well-written, and results are strong given the simplicity of the model. At the same time, multiple concerns were raised, specifically pointing out limitations of the model ( e.g., inability to handle multiple resolutions and tasks such as detection/segmentation; the model is data-hungry, parameter-inefficient, and tailored to specialized hardware/TPUs). During the discussion phase, two reviewers were disappointed by the author response, which did not address well questions such as reporting ImageNet results from other architectures, adopting the same training procedure as in DeiT, and over claimed significance of results. They downgraded their rating from 7 to 6, still recommending borderline acceptance. While the concerns raised by the reviewers are all legitimate, the AC considers that the strengths of the paper outweighs its weaknesses, and agrees with the majority that it passes the acceptance bar of NeurIPS. In particular, the proposed architecture has the potential to make an impact and inspire further research in the field, as it is conceptually simple yet achieves strong results. The authors should include in the camera-ready the discussion based on reviewer’s comments, properly convey the limitations of the architecture, and also address the two ethical concerns pointed out in the ethics review.